# Periodontal Bacteria and Outcomes Following Aneurysmal Subarachnoid Hemorrhage: A Prospective Observational Analysis

**DOI:** 10.3390/biomedicines14010048

**Published:** 2025-12-25

**Authors:** Lídia Petra Pasitka, Tihamér Molnár, Edit Urbán, Péter Csécsei, Zsolt Hetesi, Jordána Mód, Ágnes Bán

**Affiliations:** 1Department of Dentistry, Oral and Maxillofacial Surgery, Medical School, University of Pécs, 7624 Pecs, Hungary; pasitka.lidia@pte.hu (L.P.P.); ban.agnes@pte.hu (Á.B.); 2Neurocritical Care Unit, Department of Anesthesiology and Intensive Care, Medical School, University of Pécs, 7624 Pecs, Hungary; molnar.tihamer@pte.hu (T.M.); mod.jordana@pte.hu (J.M.); 3Department of Microbiology, Medical School, University of Pécs, 7624 Pecs, Hungary; 4Department of Neurosurgery, Medical School, University of Pécs, 7624 Pecs, Hungary; csecsei.peter@pte.hu; 5Institute of Mathematics and Informatics, Faculty of Natural Sciences, University of Pécs, 7624 Pecs, Hungary; hetesizs@gamma.ttk.pte.hu

**Keywords:** periodontitis, subarachnoid bleeding, outcome

## Abstract

**Background:** Periodontitis has been associated with systemic diseases such as cerebrovascular events. Emerging research highlights the potential role of the microbiome in intracranial aneurysm formation and rupture. **Aims:** We aimed to explore the associations among periodontal pathogens and the outcomes in patients with aneurysmal subarachnoid hemorrhage (aSAH). **Materials and Methods:** A total of 43 aSAH patients were enrolled. Clinical probing depth measurement and microbiological culture were performed for all participants. The markers of systemic immune response (IL-6, hsCRP) and brain injury (NSE, S100B) were measured between 24 and 48 h after admission. Development of delayed cerebral ischemia (DCI) as the primary and clinical outcome, based on modified Rankin Scale as secondary endpoints, comprised the chosen metrics. **Results:** A significant association was observed between patients with periodontal pocket depth PPD ≥ 5 mm (n = 28) and DCI, which developed in 19 patients (*p* = 0.007). In the subgroup of patients with PPD ≥ 5 mm significant associations were found between certain periodontal pathogens and DCI. Higher hsCRP (*p* = 0.05), IL-6 (*p* = 0.037) levels were observed in cases with periodontal pathogens, independent of the depth of the pocket, suggesting systemic inflammation. **Conclusions:** Elevated hsCRP and IL-6 levels, periodontal pocket depth ≥ 5 mm, and red-complex periodontal pathogens are associated with an increased risk of DCI after aSAH, suggesting a role for periodontal disease–related systemic inflammation in DCI risk stratification.

## 1. Introduction

Intracranial aneurysms (IAs) are relatively common vascular abnormalities, with an estimated prevalence of 2–3% in the general population [1]. They arise as pathological dilatations of cerebral arteries driven by hemodynamic stress and inflammation-mediated vascular wall remodeling [2]. While mechanical forces and inflammatory degradation of the aneurysm wall remain central to IA pathogenesis, emerging evidence indicates that chronic low-grade systemic inflammation may also contribute to IA formation and rupture risk. In this context, periodontitis has gained attention: several studies report an association between periodontal disease and increased risk of aneurysmal subarachnoid hemorrhage (aSAH) [3]. Supporting these findings, bacterial DNA from oral pathogens has been detected in both ruptured and unruptured IA wall specimens [4].

Periodontitis is a prevalent chronic infectious disease characterized by microbial dysbiosis and sustained low-grade inflammation of the periodontal tissues [5,6]. Beyond local tissue destruction, periodontitis is now recognized as a potential contributor to systemic disease, with links reported to more than 50 systemic conditions [7]. Host immune and genetic factors also modulate susceptibility and disease progression [8]. Among systemic consequences, vascular inflammation and endothelial dysfunction (ED) are of particular interest. Vascular endothelial cells (VECs) regulate key aspects of vascular homeostasis, including leukocyte adhesion, thrombosis, smooth-muscle-cell behavior, and inflammatory signaling [9]. When exposed to noxious stimuli, VECs may shift to a pro-inflammatory and pro-thrombotic phenotype, contributing to ED—a central mechanism implicated in atherosclerosis, diabetes, hypertension, and neuroinflammatory disorders [10,11]. Although epidemiological and clinical data suggest that periodontitis is associated with impaired endothelial function [12], the underlying mechanisms—including direct microbial invasion, systemic cytokine release, and alterations in vascular permeability—remain incompletely understood [13].

Periodontitis is a common condition associated with a chronic low-grade inflammatory state, and it contributes to the development of atherosclerotic vascular disease [14]. In addition, a growing body of evidence highlights oxidative stress as an additional systemic consequence of periodontal infection, with reference to cerebrovascular and neurological pathology [15]. Excess reactive oxygen species (ROS) generated during periodontal inflammation contribute to systemic redox imbalance, impair endothelial nitric oxide bioavailability, and promote endothelial dysfunction and vascular stiffness [16]. Oxidative stress can also disrupt blood–brain barrier integrity and activate redox-dependent inflammatory pathways, thereby facilitating microglial activation and neural injury. Thus, oxidative stress represents an important mechanistic link between oral pathogens, vascular damage, and neuroinflammation that extends beyond direct bacterial effects.

Several periodontopathogens, including *Porphyromonas gingivalis*, *Tannerella forsythia*, *Treponema denticola*, and *Parvimonas micra*, exhibit virulence traits that promote immune evasion and may facilitate dissemination beyond the oral cavity [17,18]. *P. gingivalis* in particular can invade endothelial cells, induce oxidative stress, and activate NF-κB–mediated pro-inflammatory cytokine production (e.g., IL-1β, TNF-α) [19,20]. Experimental studies also indicate that periodontopathogens can alter vascular permeability via Mfsd2a/Caveolin-1-dependent pathways and modulate the complement system, potentially sustaining local vascular inflammation. Correspondingly, oral bacterial DNA and proteins have been detected in IA walls, where they may trigger macrophage activation, collagen degradation, and inflammatory vascular remodeling. Clinical findings further support associations between elevated periodontal pathogen burden and both ischemic and hemorrhagic stroke, including aSAH [18].

Systemic effects of periodontitis extend to neurological pathways as well. Animal models demonstrate that periodontal inflammation can promote atherosclerosis through endothelial-to-mesenchymal transition [20,21], and similar mechanisms have been observed in patients with lacunar infarction [22]. The relationship between neurological disorders and periodontitis is bidirectional: neurodegenerative or motor impairments can worsen oral hygiene, while inflammatory mediators released from periodontal lesions may disrupt the blood–brain barrier, facilitate microglial activation, and promote neuroinflammation [23,24,25,26,27].

Delayed cerebral ischemia (DCI) remains a major determinant of poor outcome after aSAH. Although mechanisms such as microthrombosis, neuroinflammation, disordered coagulation, impaired fibrinolysis, cortical spreading depolarizations, and immune activation have been implicated [28], the potential contribution of periodontal pathogens to DCI is largely unexplored. Recent studies reporting an increased risk of IA rupture in the presence of *P. gingivalis* and *Aggregatibacter actinomycetemcomitans* suggest that dysregulated immune responses to oral pathogens may influence aneurysm instability, but their role in DCI development has not yet been clarified [28].

Markers of early brain injury, such as S100B (glial injury) and neuron-specific enolase (NSE), correlate with disease severity and can predict vasospasm and neurological outcome after aSAH [29,30]. These biomarkers, together with systemic inflammatory markers such as IL-6 and high-sensitivity C-reactive protein (hsCRP), may help elucidate interactions between periodontal disease and cerebrovascular injury.

The primary aim of this prospective clinical study was to investigate the relationships between the severity of periodontitis, the presence of specific periodontal pathogens, and the occurrence of DCI in patients with aSAH. A secondary objective was to evaluate the prognostic and predictive potential of periodontal bacterial profiles in relation to clinical outcomes, systemic inflammatory markers (IL-6, hsCRP), and biomarkers of brain injury (S100B, NSE).

## 2. Materials and Methods

### 2.1. Study Design and Population

The institutional review board approval was obtained previously (9596-PTE2023). Written informed consent was obtained from patients or their legal representatives prior to their inclusion in the study. A total of 43 patients diagnosed with aSAH at our institution from February 2023 to November 2024 and treated by coiling within 2 days after admission were prospectively included. Our study’s inclusion criteria were as follows: 1. age  >  18 years old; 2. confirmed diagnosis of aSAH by non-contrast head CT, and established diagnosis of aneurysm by CTA or DSA; and 3. diagnosis occurred within 24 h after index event.

All patients underwent standardized neuroimaging follow-up according to institutional aSAH protocols. Non-contrast cranial CT was performed on admission and repeated as clinically indicated. Daily transcranial Doppler (TCD) examinations were used to screen for cerebral vasospasm; elevations in mean flow velocity or Lindegaard ratio triggered confirmatory imaging. Any acute neurological deterioration, including new focal deficits or a sustained decrease in level of consciousness, prompted immediate neuroimaging. The first-line modality was non-contrast CT, followed—when appropriate—by CT angiography (CTA), CT perfusion (CTP), or MR or digital subtraction angiography (DSA) to evaluate for vasospasm and new ischemic injury.

The exclusion criteria were as follows: traumatic SAH, pregnancy, hospital admission later than 24 h after the index event, lack of aneurysm treatment, bleeding from arteriovenous malformation, absence of a signed consent form, underlying systemic diseases (malignancies or liver and/or renal insufficiency, and chronic lung disease, inflammatory bowel disease, or any known chronic gastrointestinal diseases), chronic infection, or signs of any acute infection on admission. For eligible patients, their medical records, including age, gender, risk factors (hypertension, diabetes, and smoking status), blood pressure and laboratory data on admission, baseline WFNS and mFisher score, interventions performed during intensive care (extra-ventricular drain, lumbar drain, etc.), presence of DCI, and clinical outcomes (modified Rankin Scale, mRS) were reviewed and recorded. All patients received nimodipine 6 times 60 mg per orally from the first day for vasospasm prevention [31].

The diagnosis of DCI was made only after thoroughly ruling out other potential causes and required the consensus of at least two experts in intensive neurological medical care. DCI was diagnosed according to the Vergouwen et al. criteria, requiring both (i) a new focal neurological impairment or decrease in consciousness unexplained by rebleeding, hydrocephalus, seizures, metabolic disturbances, or procedural complications; and (ii) radiological evidence of new cerebral ischemia on CT, CTP, CTA, or DSA [32]. These imaging criteria and diagnostic procedures were applied uniformly across all study participants.

The criteria for defining systemic and central nervous system infection included the presence of infection symptoms with fever (>38 °C) and elevated levels of C-reactive protein (rising CRP level after an initial peak) and/or procalcitonin (>0.5 ng/mL), combined with a positive diagnostic test, such as a chest X-ray, CSF and/or blood culture, or urine analysis.

The control group consisted of otherwise healthy individuals undergoing routine dental screening, without periodontitis, matched by age ±5 years and gender. A history of any infectious or central nervous system diseases was an exclusion criterion for this group.

### 2.2. Periodontal Probing

The assessment of the dental status was performed according to the WHO guidelines [13]. All periodontal examinations were performed by a single calibrated examiner, which eliminated inter-examiner variability. Full-mouth probing was not feasible due to ICU constraints, (e.g., limited patient cooperation, sedation or intubation), and the need to minimize procedure time for safety, while acknowledging potential misclassification in critically ill patients. Therefore, the examination was conducted only on the Ramfjord teeth as a surrogate for full-mouth assessments [33]. “Ramfjord’s Teeth” refers to six specific index teeth (16, 21, 24, 36, 41, and 44) strategically selected to represent the overall condition of the mouth. Regarding missing Ramfjord teeth, when a designated Ramfjord tooth was absent, the corresponding contralateral tooth was examined. The data collected from these teeth can be shared with patients to support behavior change initiatives and serve as a reliable, long-term evaluation tool for the dental team [33].

The depth of the periodontal pocket was determined using a calibrated periodontal probe. During probing, a pressure of 0.2 N/mm^2^ was applied [34]. The probe was inserted into the sulcus until resistance was felt, and then the examiner gently explored the base of the sulcus at 1 mm intervals without lifting the probe out of the sulcus. During the examination, the examiner determined the sulcus depth next to the teeth (PPD = clinical probing depth) and the clinical attachment loss (CAL = recession + PPD). For each tooth, the 6 deepest values were recorded based on the mesio-, centro-, and disto-oral surfaces and the mesi-, centro-, and disto-vestibular surfaces. We recorded the presence of biofilm, calculus, pus, bleeding, and recession associated with the surfaces, and examined the mobility of the teeth. For multi-rooted teeth, the Nabers probe was used to determine horizontal furcation involvement. This painless procedure did not cause any additional psychological or physical strain for the patient.

### 2.3. Microbiological Sampling and Sample Culturing

Subgingival plaque samples were obtained from the three deepest pockets, or most diseased sites, with three individual sterile paper points, which were placed in the gingival crevice for 15 s and moved around the abutment and then sent to the laboratory in Portagerm multitransport medium [35]. All cultures were commenced within 1 h of sampling.

The samples (three paper points/patient) were immediately sent to the microbiology laboratory, where all three paper points were placed together into 1.0 mL reduced brain–heart infusion (BHI pH 7.2) and mixed on a Vortex shaker for 30 s. After gentle dispersion, the suspensions were diluted (10-1–10-6) in reduced BHI broth. Then, 100–100 µL of the stock solution itself and of each dilution were plated in parallel on selective and non-selective agar media. Schaedler agar (containing horse blood 5% *v*/*v*, haemin, and vitamin K1; bioMérieux, Marcy l’Etoile, France) supplemented with 5% (*v*/*v*) bovine blood, haemin, and vitamin K1 was used to isolate and count all culturable anaerobic bacteria. Columbia blood and boiled blood (chocolate) media were used for the isolation and counting of aerobic/microaerophilic bacterial flora. Eosine-methilene blue medium (bioMérieux, S.A., Marcy l’Etoile, France) was used for the selective cultivation of potentially occurring aerobic Gram-negative *Enterobacterales* species and non-fermentative Gram-negative bacteria, while Sabouroud Dextrose medium was used for the cultivation of fungi. On a solid plate, cultures were incubated for 5 days in an anaerobic chamber (Bactron, Sheldon Manufacturing, Cornelius, OR, USA) in an anaerobic atmosphere (90% N_2_, 5% H_2_, and 5% CO_2_), or in a CO_2_ atmosphere for 6 days at 37 °C. For aerobic bacteria, the plates were cultured at 37 °C in a 5% CO_2_-containing environment for 48 h. The selective agar media for the isolation of *Enterobacterales* and non-fermentative Gram-negative bacteria were incubated at 37 °C for 24 h. Fungi were incubated at 37 °C in ambient air for 24 h and at room temperature for a further 5 days. Each different colony type from positive cultures was subcultured for purity and identification. The results from the Gram staining and the atmospheric growth requirements of each colony type were used to identify the isolates.

The cultured bacteria and fungi were counted by determining the exact number of colonies, and the strains with different colony morphology were identified at the species level using the Matrix-assisted Laser Desorption, Ionization, and Time-of-Flight mass spectrometry (MALDI-TOF MS; Bruker Daltonik, Bremen, Germany) identification method [36]. (Sample preparation procedures and the technical details of the MALDI measurements have been described in detail elsewhere.) The microFlex LT Biotyper instrument (Bruker Daltonik, Bremen, Germany), the MALDI Biotyper RTC 3.1 software (Bruker Daltonik, Bremen, Germany), and the MALDI Biotyper Library 3.1 were used for analyses of the spectra. Based on consensus criteria, genus-level identification of bacteria isolates was deemed reliable if the log(score) ≥ 1.7, while for reliable species-level identification, a log(score) ≥ 2.0 was needed.

### 2.4. Biomarkers

Serum levels of hsCRP, Il-6, S100, and NSE were measured from patient samples collected between 24 and 48 h after the aSAH event, before any clinical infection occurred. Blood samples were dispensed into closed-system serum separator tubes with the anticoagulant Vacuette^®^ (Greiner Bio-One GmbH, Kremsmünster, Austria) from arterial cannulae routinely inserted for blood pressure monitoring and blood sampling. After blood collection samples were immediately transported to the laboratory and centrifuged at room temperature for 10 min at 1500× *g*. After centrifugation, the measurements were performed immediately. The hsCRP was measured from patients’ sera with Tina-quant^®^ C-Reactive Protein IV reagent (Roche Hungary Ltd., Budapest, Hungary) on Roche Cobas 6000 fully automated Chemistry Analyzer (Roche Diagnostics, Basel, Switzerland). This test is a two-reagent immunoturbidimetric assay based on particle-enhanced immunological agglutination, and its reference range in adult serum is <5 mg/L. For measurement of IL-6, S100, and NSE we used Roche Elecsys^®^ IL-6, Roche Elecsys^®^ NSE, and Roche Elecsys^®^ S100 (Roche Hungary Ltd.) reagents on a fully automated Cobas e801 analyzer (Roche Diagnostics) [37]. All three tests use the electrochemiluminescence immunoassay (ECLIA) technique, which allows the in vitro qualitative detection of analytes in the sample. For Elecsys IL-6 test the cut-off value is ≤7 pg/mL. The cut-off value provided by the manufacturer for Elecsys NSE assay is ≤16.3 µg/L. The Roche Elecsys^®^ S100 immunochemical assay is suitable for the in vitro quantification of S100 A1B and S100 BB in human serum. Serum S100 values ≤ 0.105 µg/L indicate the absence of intracranial injury/bleeding.

### 2.5. Outcome Measures

The primary outcome was the diagnosis of DCI based on clinical (decline of GCS ≥ 2 points or de novo neurological deficit), neuroimaging (novel DWI lesion) and angiography (signs of cerebral vasopsasm) findings, while modified Rankin Scale (mRS) at discharge from the ICU was used as secondary outcome. Favorable outcome was defined as an mRS of 0–2, whereas poor outcome was defined as a value of 3–6.

### 2.6. Statistical Analysis

The data were analyzed using SPSS 25.0 (SPSS Statistics v22.0; IBM Corp., Armonk, NY, USA). Qualitative variables were represented as frequencies (percentages), while normally and non-normally distributed continuous variables were presented as means (standard deviations, SD) and medians (interquartile ranges), respectively. Comparisons of different variables between the two groups were conducted using the Chi-square test or Fisher’s exact test for qualitative data, and the Mann–Whitney U test or independent t-test for quantitative data. The Spearman correlation coefficient was used to calculate the bivariate correlation between serum NSE, S100B, hsCRP, and IL-6 levels. A binary logistic regression analysis was used to find independent predictors of DCI. Group differences in mean log_10_ (CFU/mL) were tested with a two-sided Welch’s t-test (unequal variances. To avoid overfitting, we used a bias-reduced logistic model, because the sample was small and some 2 × 2 cells are sparse; standard logistic regression can yield unstable or inflated odds ratios. Firth’s penalized likelihood corrects small-sample bias and stabilizes the point estimate and its confidence interval, while answering the same clinical question.

### 2.7. Ethics Approval

This study protocol was approved by the Scientific and Research Ethics Committee of the Medical Research Council of the University of Pécs (approval number: 9596-PTE2023 approved on 31 March 2023). Written informed consent was obtained from each participant in accordance with institutional and national ethics regulations.

## 3. Results

### 3.1. Patient Characteristics and Control Group

A total of 43 aSAH patients and 10 healthy control subjects were enrolled in this prospective study. The mean age of the patient population was 57.4 ± 12.5 years, and 56.6 ± 12.6 years in the control group. The gender distribution was also similar in the patient (70% females) and control groups (60% females). Fewer smokers were found among the control subjects (20%). In the control group, the median values (interquartile range) for hsCRP, IL-6, NSE, and S100B were as follows: 3.2 (2.0–3.5) mg/L, 1.5 (1.5–1.65) ng/L, 11.1 (10.2–12.6) µg/L, and 0.046 (0.03–0.07) µg/L, respectively. All markers but the S100B level were significantly lower in the healthy control group (for all, *p* < 0.05). The clinical characteristics of the aSAH patients involved as well as the favorable- vs. poor-outcome groups after mRS-based dichotomization are summarized in Table 1. Demography and comorbidities such as hypertension, diabetes, and smoking were similar in the two mRS-based outcome groups. The clinical severity (WFNS) and neuroimaging (mFischer score) on admission were significantly worse in the poor-outcome group. Similarly, the systemic concentration of the thrombo-inflammatory markers such as IL-6 and CRP were significantly higher in the poor-outcome group compared to the favorable group. Serum CRP level with a cut-off > 32 mg/L (sensitivity: 75%; specificity: 99%; area: 0.904, 95% CI: 0.770–1.039, *p* < 0.001), and serum concentration of IL-6 with a cut-off > 17 pg/mL (sensitivity: 87.5%; specificity: 94%; area: 0.912, 95% CI: 0.778–1.046, *p* < 0.001) predicted a poor outcome (mRS ≥ 3) at discharge from the ICU (Figure 1). Interestingly, the neuronal injury marker NSE showed no significant difference, but the astrocyte specific S100B showed a significant difference in comparison between the favorable- and poor-outcome groups. Regarding the secondary outcome, significantly more DCI developed among patients with poor outcome and the death rate was higher in this group as well.

### 3.2. Delayed Cerebral Ischemia

Demographic and clinical characteristics of aSAH population, dichotomized by the presence or absence of DCI are shown in Table 2. As a tendency, the mFischer score was higher, reflecting more blood in the subarachnoidal space in those patients in whom DCI developed, predominantly on the days 7–10 post-aSAH. Importantly, the binary logistic regression analysis, including covariates such as age, gender, and all biomarkers, revealed that the presence of PPD ≥ 5 mm is an independent predictor of DCI (OR: 124; 95% CI: 1.4–11,359, *p* = 0.03). Using the ROC analysis, the combination of serum concentrations of CRP and IL-6 with PPD > 5 mm was proved to be a strong predictor of DCI during ICU stay (area: 0.814, 95% CI: 0.638–0.990, *p* < 0.001) (Figure 2a). Moreover, the best combination for predicting DCI comprised the serum levels of hsCRP and IL-6 and the culture-confirmed presence of *Fusobacterium (F.) nucleatum* in periodontal pockets deeper than 5 mm (Figure 2b). In addition to the correlation found for *F. nucleatum*, the presence of *Parvimonas micra* (n = 14 cases) also showed a correlation with DCI (*p* = 0.018).

### 3.3. Biomarkers and Periodontitis

The systemic concentration of the thrombo-inflammatory markers (IL-6 and hsCRP) were significantly higher in patients with PPD ≥ 5 mm compared to those with PPD < 5 mm (Table 3). Interestingly, in contrast to the astrocyte-specific S100B, the neuronal injury marker NSE was found to be significantly higher in the PPD ≥ 5 mm group. In particular, the presence of certain bacteria was associated with an increased level of NSE in patients with periodontitis (see below, in the discussion of deep pockets and periodontal pathogens).

#### 3.3.1. Inflammatory Markers and Periodontal Pathogens

Higher serum concentration of IL-6 (*p* = 0.037) was observed in cases with periodontal bacteria, but it showed no association with the depth of the pockets, suggesting that other factors may also contribute to the magnitude of systemic inflammation. Each isolated bacterium was also quantified by colony-forming unit (CFU)/mL. However, the maximal value of CFU/mL showed no correlation with the systemic level of the inflammatory markers. Taken together, systemic inflammation may not be driven solely by the bacterial quantity or the severity of local periodontal destruction. Instead, this points toward a more complex interplay of host immune response and microbial virulence factors, and possibly, systemic susceptibility.

#### 3.3.2. Deeper Pockets and Periodontal Pathogens

The common bacteria in periodontal pockets of patients with aSAH were as follows: *Porphyromonas gingivalis* (colony-forming unit/mL: CFU/mL max: 10^7^), *Prevotella denticola* (CFU/mL max: 10^6^), *Porphyromonas asaccharolytica* (CFU/mL max: 10^5^), *Porphyromonas somerae* (CFU/mL max: 10^7^); *Fusobacterium nucleatum* (CFU/mL max: 10^7^), *Fusobacterium periodonticum* (CFU/mL max: 10^6^), *Prevotella intermedia* (CFU/mL max: 10^7^), *Prevotella oris* (CFU/mL max: 10^5^), *Prevotella melaninogenica* (CFU/mL max: 10^7^), *Prevotella denticola* (CFU/mL max: 10^6^), *Prevotella nigrescens* (CFU/mL max: 10^6^), *Parvimonas micra* (CFU/mL max: 10^7^), *Leptotrichia buccalis* (CFU/mL max: 10^5^), and *Veillonella parvula* (CFU/mL max: 10^6^). Table 4 presents the minimum and maximum bacterial concentrations (log10 CFU), the mean values, and the statistical significance of differences between patients with DCI as a post-SAH complication and those without DCI (uncomplicated cases). In our cohort, the mean CFU value of *Veillonella parvula* was significantly higher among patients with DCI. Moreover, the overall prevalence of all periodontopathogenic bacteria listed in Table 4 was higher in the DCI group, compared to subjects without DCI. There was a positive correlation between depth of periodontal pockets, expressed in mm, and hsCRP (0.398, *p* = 0.013) as well as CFU (0.489, *p* < 0.001) in the total aSAH population. In patients with PPD ≥ 5 mm, DCI positively correlated with a poor outcome, as measured by mRS (*p* = 0.016). Periodontal assessment was also performed in the healthy control group, and these data are provided in Appendix A. Periodontal examination and sampling in controls followed the same protocol as in the aSAH patients, enabling comparison of periodontal status. However, detailed oral hygiene–related variables were not systematically collected in controls and were therefore not included in the analyses.

#### 3.3.3. Findings with the Bias-Reduced Model

(i)DCI as outcome; periodontal pocket depth (PPD ≥ 5 mm) as exposure.

Using Firth penalized logistic regression, the estimated odds ratio for DCI associated with PPD ≥ 5 mm was 7.00 (95% CI: 1.40–34.97). The association was statistically supported by the likelihood-ratio test (*p* = 0.004). Notably, this analysis corrects for sparse-cell bias arising from the small number of DCI events among patients without deep periodontal pockets (n = 2), and the magnitude and direction of the association were preserved after penalization.

(ii)Poor functional (mRS: 3–6) outcome as outcome; periodontal pocket depth (PPD ≥ 5 mm) as exposure.

The bias-reduced estimate for poor functional outcome yielded an odds ratio of 2.27 (95% CI: 0.53–9.80), with *p* = 0.147 by the likelihood-ratio test. Although the point estimate suggests higher odds of poor outcome in patients with deeper periodontal pockets, the confidence interval is wide and includes the null, indicating substantial imprecision and insufficient statistical support in this sample.

**Table 4 biomedicines-14-00048-t004:** CFU of bacteria isolated from the periodontal pockets in patients with and without DCI.

	Without DCI n = 24	With DCI n = 19	*p*
*Veillonella parvula* (min–max)	n = 20; 83% 4.95 ± 0.826 (4–7)	n = 16; 84% 5.812 ± 0.981 (4–7)	0.009
*Fusobacterium nucleatum* (min–max)	n = 9; 37% 5.538 ± 0.877 (4–7)	n = 13; 68% 6 ± 0.866 (5–7)	0.238
*Parvimonas micra* (min–max)	n = 2; 8% 5.5 ± 0.707 (5–6)	n = 10; 53% 5.8 ± 0.632 (5–7)	0.656
*Porphyromonas gingivalis* (min–max)	n = 4; 16% 6 ± 0	n = 8; 42% 5.875 ± 0.835 (5–7)	0.685
*Prevotella intermedia* (min–max)	n = 2; 8% 5 ± 0	n = 4; 21% 5.25 ± 0.5 (5–6)	0.391

CFU, colony-forming units; DCI, delayed cerebral ischemia. Data are presented as the absolute number of patients positive for each bacterial species (n), the corresponding occurrence rate (% of patients in each group), and the mean ± SD log_10_(CFU/mL), with minimum–maximum values. Group differences in mean log_10_(CFU/mL) were assessed using a two-sided Welch’s *t*-test (unequal variances). *p*-values indicate differences between patients with and without DCI.

## 4. Discussion

Periodontitis is a common chronic biofilm-associated inflammatory disease characterized by dysbiotic microbial communities and progressive destruction of tooth-supporting tissues. Beyond local pathology, periodontal inflammation has been linked to systemic conditions—including cardiovascular and cerebrovascular disease—through mechanisms involving transient bacteremia, sustained cytokine release, endothelial dysfunction, and oxidative stress [15,38]. A limited group of keystone pathogens (e.g., *Porphyromonas gingivalis*, *Tannerella forsythia*, and *Treponema denticola*) can disproportionately amplify host inflammatory and redox-regulated responses, potentially affecting vascular and neurological integrity. These pathways are biologically relevant in aneurysmal subarachnoid hemorrhage, in which oxidative stress and inflammation contribute to secondary brain injury and delayed cerebral ischemia. Accordingly, we examined associations between periodontal status, systemic inflammatory and brain injury biomarkers, and clinical outcomes after aSAH, while minimizing confounding by excluding patients with acute infections or immunosuppression.

### 4.1. Clinical Severity and Outcome Determinants

In our cohort of 43 patients, well-established predictors of outcome after aSAH—such as WFNS grade and modified Fisher score—were significantly worse among patients with poor neurological outcomes (mRS ≥ 3). Notably, early increases in IL-6 and hsCRP were also associated with a poor outcome, underscoring the role of systemic inflammatory burden in exacerbating secondary brain injury [39,40]. Because IL-6 and hsCRP rise rapidly within hours of an inflammatory stimulus, their early elevation likely reflects the combined effect of the hemorrhage and any pre-existing inflammatory conditions, including chronic periodontitis. Interestingly, while NSE did not differ significantly between outcome groups, S100B was markedly elevated in patients with poor outcomes. This suggests a stronger contribution of astroglial injury to early global brain damage after aSAH—consistent with observations in traumatic and ischemic brain injury [41]. Importantly, S100B typically shows delayed elevation after aSAH, often peaking several days after hemorrhage as the blood–brain barrier disruption and secondary injury processes progress. Because our sampling occurred at a single, early time point (24–48 h), interpretation of S100B dynamics must consider its delayed kinetic profile; later increases related to DCI could not be captured without serial sampling.

### 4.2. Delayed Cerebral Ischemia and Periodontal Inflammation

DCI occurred in 44% of patients and was strongly associated with both poor neurological outcomes and increased ICU mortality. Patients who developed DCI had a higher prevalence of severe periodontitis (PPD ≥ 5 mm), suggesting that chronic oral inflammatory burden may contribute to cerebrovascular vulnerability during the post-aSAH period. Bias-reduced logistic regression yielded results that were directionally consistent with the primary analyses and mitigated concerns about small-sample and sparse-data bias. The association between deep periodontal pockets (PPD ≥ 5 mm) and DCI remained moderate-to-strong in magnitude after correction, supporting the robustness of this exploratory signal. Accordingly, we emphasize effect sizes and confidence intervals rather than dichotomous significance and interpret these findings as hypothesis-generating, warranting confirmation in larger, adequately powered cohorts. Next, ROC analysis further showed that combining periodontal status with systemic inflammatory markers (IL-6 and hsCRP) significantly improved prediction of DCI. The most informative combination included hsCRP, IL-6, and *Fusobacterium nucleatum*, a pathogen increasingly linked to systemic vascular inflammation, endothelial dysfunction, and cerebrovascular pathology [42,43]. These findings integrate with growing evidence that oral dysbiosis may influence cerebral small-vessel injury and white-matter lesions, even in individuals without overt neurological disease [44].

### 4.3. Periodontitis, Inflammatory Burden, and Brain Injury

Patients with PPD ≥ 5 mm showed significantly elevated IL-6 and hsCRP, reinforcing the systemic inflammatory nature of chronic periodontal disease [45,46]. Notably, NSE—but not S100B—was significantly higher in patients with severe periodontitis. Given the rapid release of NSE following neuronal injury and the slower, more delayed kinetics of S100B, this pattern suggests that periodontitis-related systemic inflammation may preferentially exacerbate early neuronal stress, rather than immediate astrocytic damage. Mechanisms may include cytokine-mediated neurotoxicity, increased permeability of inflamed mucosal tissues allowing bacterial translocation, or oxidative-stress–driven endothelial dysfunction affecting cerebral microcirculation [15,47].

Moreover, the presence of higher loads of bacteria in the subgingival biofilm—especially *Veillonella*, *Fusobacterium nucleatum*, *Parvimonas micra*, and combinations involving *Porphyromonas* and *Prevotella* species—was associated with the occurrence of delayed cerebral ischemia (DCI). While direct evidence linking these specific taxa to DCI in aSAH patients remains limited, previous studies have reported the systemic pro-inflammatory effects of these periodontal pathogens, which may exacerbate neuroinflammation and vascular dysfunction post-aSAH. For instance, *F. nucleatum* and *P. gingivalis* have been shown to contribute to endothelial activation and disruption of the blood–brain barrier, mechanisms potentially involved in the pathogenesis of DCI. These bacteria are known for their potent pro-inflammatory properties and ability to penetrate vascular tissues [48,49]. Their detection in patients with deeper periodontal pockets further strengthens the proposed link between oral dysbiosis and cerebrovascular complications in aSAH patients.

### 4.4. Pathophysiological Implications and Clinical Relevance

The interplay between systemic inflammation, periodontitis, and DCI suggests a plausible pathophysiological cascade. Chronic periodontal inflammation elevates systemic inflammatory mediators, primes the vascular endothelium, and potentially facilitates bacterial dissemination, as the aortic inflammatory response induced by polymicrobial infection with well-characterized periodontal pathogens has recently been demonstrated [50]. In the vulnerable milieu following aSAH, this inflammatory burden may exacerbate cerebrovascular dysregulation, contributing to the development of DCI and worsening neurological outcomes [51].

These findings may have important clinical implications. First, periodontal screening and management might become a relevant adjunct in the holistic care of aSAH patients, particularly those at high risk of DCI. Second, systemic inflammation markers such as IL-6 and CRP, in combination with oral health assessment, could refine risk stratification models for ICU monitoring and intervention planning [52].

### 4.5. Limitations and Future Directions

Several limitations must be acknowledged. The relatively small sample size limits the generalizability of our findings and contributes to wide confidence intervals in the regression models; accordingly, the statistical analyses should be interpreted as exploratory. The observational nature of this single-center study precludes causal inference, and residual confounding cannot be excluded. Methodologically, full-mouth periodontal probing was not feasible in the ICU setting due to several reasons (e.g., lack of cooperation, sedation, and intubation) in addition to the time-sensitive clinical constraints; therefore, examination of Ramfjord teeth was used as a validated surrogate, although some misclassification of periodontal severity remains possible. Furthermore, while the presence of periodontal pathogens was confirmed using culture-based microbiology with MALDI-TOF identification, this approach may under-detect strict anaerobes and low-abundance taxa, and CFU values reflect only cultivable organisms rather than total microbial load. Biomarker sampling occurred within the first 24–48 h after aSAH rather than at immediate baseline, during a period of physiologic variability, and although samples were collected prior to any clinically documented infections, this timing may still affect interpretation of early inflammatory markers. Finally, the modest number of outcome events limited the number of covariates that could be included in regression models, raising the possibility of overfitting, despite the use of penalized methods.

Future studies should validate these findings in larger, multicenter cohorts and further delineate mechanistic pathways, including molecular characterizations of the oral microbiome, biofilm structure, microbial virulence determinants, bacterial translocation, endothelial dysfunction, and neuroinflammatory responses associated with DCI. Interventional trials assessing whether periodontal treatment can modulate systemic inflammation and reduce DCI risk in aSAH patients would be particularly valuable. In addition, investigation of the interplay between oral and intestinal dysbiosis in critically ill patients may yield further insight into host–microbiome interactions influencing post-SAH outcomes [53].

## 5. Conclusions

This study may support an association between periodontitis and adverse neurological outcomes in aSAH, mediated by elevated systemic inflammation and increased risk of DCI. Oral health status, particularly the presence of deep periodontal pockets and some specific pathogenic bacteria, may represent a modifiable risk factor in the complex management of aSAH patients [54].

Taken together, these findings indicate that monitoring of serum inflammatory markers such as hsCRP and IL-6, along with PPD deeper than 5 mm and red-complex bacterial pathogens may support early detection of patients at risk of DCI after aSAH. These results need to be validated in larger prospective cohorts.

## Figures and Tables

**Figure 1 biomedicines-14-00048-f001:**
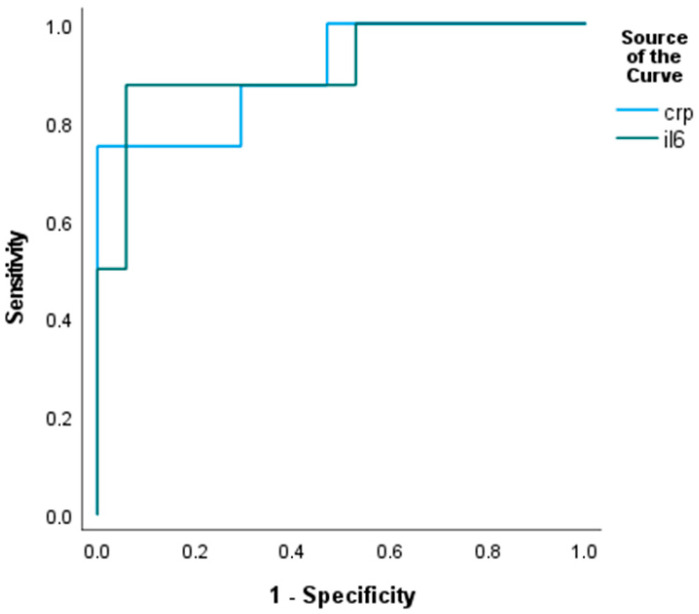
ROC analysis of serum CRP and IL-6 levels relative to prediction of poor outcome.

**Figure 2 biomedicines-14-00048-f002:**
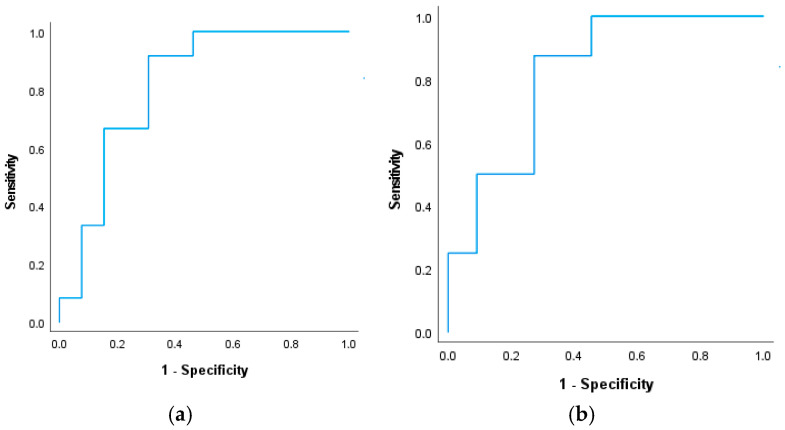
ROC curves for prediction of delayed cerebral ischemia: (**a**) predicted probability of the combination of serum concentrations of CRP and IL-6 with PPD > 5 mm; (**b**) predicted probability when CRP, IL-6, PPD > 5 mm was combined with confirmed *F. nucleatum* in the culture taken from the periodontal pocket.

**Table 1 biomedicines-14-00048-t001:** Demographic and clinical characteristics of aSAH population, dichotomized by mRS-based outcome.

Variable	Total (n = 43)	mRS < 3 (n = 28)	mRS ≥ 3 (n = 15)	*p*-Value
Age (mean ± SD)	57.4 ± 12.5	55.4 ± 12.1	61.3 ± 12.6	0.140
Female, N (%)	30 (70)	19 (68)	11 (73)	0.709
Hypertension, N (%)	24 (56)	14 (50)	10 (66)	0.294
Diabetes, N (%)	2 (5)	2 (7)	0 (0)	0.289
Smoking, N (%)	16 (37)	9 (32)	7 (46)	0.348
WFNS, median (IQR)	2 (1–4)	1 (1–1)	3.5 (2–5)	0.031
mFisher score, median (IQR)	3 (2–3.5)	2 (2–3)	4 (2.5–4)	0.018
PPD > 5 mm	28 (65)	16 (57)	12 (80)	0.221
Aneurysm location, N (%)				
Internal carotid artery	4 (9)	2 (7)	2 (13)	
Middle cerebral artery	12 (28)	8 (29)	4 (27)	
Anterior communicating	11 (26)	7 (25)	4 (27)	
Posterior communicating	3 (7)	2 (7)	1 (7)	
Anterior cerebral artery	2 (5)	1 (4)	1 (7)	
Vertebral and Basilar	7 (16)	4 (14)	3 (20)	
NSE, median (IQR)	8.5 (5.2–12.5)	7.9 (5.1–11.7)	12.3 (11.2–22.4)	0.123
S100B, median (IQR)	0.05 (0.02–0.10)	0.04 (0.02–0.07)	0.1 (0.05–0.21)	0.032
IL-6, pg/mL, median (IQR)	8.1 (3.5–28.1)	5.8 (3.2–10.1)	55.6 (28.1–149)	0.005
hsCRP, mg/L, median (IQR)	11.8 (4.4–31.9)	9.0 (3.3–21)	41.1 (22.4–63.9)	0.003
Lumbal drain, N (%)	17 (40)	12 (43)	5 (33)	0.803
Extra-ventricular drainage, N (%)	11 (26)	1 (3.5)	10 (66)	<0.001
Delayed cerebral ischemia, N (%)	19 (44)	6 (21)	13 (86)	<0.001
Death, N (%)	9 (21)	2 (7)	7 (46)	<0.001

SD, standard deviation; N, number; WFNS, World Federation of Neurological Societies; PPD, periodontal pocket depth; IQR, interquartile range; NSE, neuron-specific enolase; IL-6, interleukin-6; hsCRP, high-sensitivity C-reactive protein. We defined favorable outcomes as modified Rankin scores (mRS) 0–2, while unfavorable outcomes were associated with mRS 3–6. *p*-value indicates favorable vs. unfavorable comparison.

**Table 2 biomedicines-14-00048-t002:** Demographic and clinical characteristics of SAH population, dichotomized by the presence or absence of DCI.

	Total n = 43	Without DCI n = 24	With DCI n = 19	*p*
Demography				
Age, y	55.9 ± 10.8	50 ± 7.5	61.1 ± 10.9	0.101
Female (%)	30 (70)	17 (71)	13 (68)	0.833
Clinical features				
Hypertension (%)	24 (56)	14 (58)	10 (53)	0.708
DM (%)	2 (5)	2 (8)	0	0.198
Smoking (%)	16 (37)	8 (33)	8 (42)	0.555
Dyslipidaemia (%)	4 (10)	1 (4)	3 (16)	0.193
GCS	13 ± 3	14 ± 1	12 ± 4	0.183
ICU days (%)	13 ± 4	11 ± 4	14 ± 4	0.267
mRS ≥ 3 (%)	15 (35)	2 (8)	13 (68)	<0.001
ICU mortality (%)	9 (21)	2 (8)	7 (37)	0.022
PPD ≥ 5 mm (%)	28 (65)	11 (46)	17 (89)	0.007
Fischer score	2.1 ± 1.2	1.4 ± 0.8	2.6 ± 1.3	0.05

DCI, delayed cerebral ischemia; DM, diabetes mellitus; GCS, Glasgow Coma Scale; ICU, intensive care unit; mRS, modified Rankin Scale; poor outcome, mRS ≥ 3; PPD, periodontal pocket depth; *p*-value indicates without DCI vs. with DCI comparison.

**Table 3 biomedicines-14-00048-t003:** Biomarkers in the total population and as dichotomized by the presence or absence of PPD ≥ 5 mm.

	Total Patients n = 43	PPD < 5 mm n = 15	PPD ≥ 5 mm n = 28	*p*
Thrombo-inflammathory markers				
hsCRP	11.8 (4.4–31.9)	4.9 (1.0–13.5)	24.4 (11.7–41.7)	0.005
Lymphocytes	1.8 (1.3–1.9)	1.8 (1.3–2.3)	1.9 (1.4–1.9)	0.768
Neutrophils	7.2 (4.3–11.4)	5.8 (3.9–9.3)	9.2 (5.7–11.6)	0.315
IL-6	8.1 (3.5–28.1)	3.7 (2.9–6.8)	11.0 (5.3–55.9)	0.031
NLR	3.7 (2.8–6.5)	2.7 (1.4–5.5)	4.7 (3.1–7.4)	0.195
Platelets	268 (216–298)	235 (206–280)	280 (263–348)	0.223
Brain injury markers				
NSE	9.5 (6.5–13.2)	5.1 (2.2–9.0)	11.2 (8.0–14.0)	0.013
S100B	0.03 (0.05–0.09)	0.04 (0.02–0.07)	0.06 (0.04–0.11)	0.168

PPD, periodontal pocket depth; hsCRP, high-sensitivity C-reactive protein; IL-6, interleukin-6; NLR: neutrophil–lymphocyte ratio; NSE, neuron-specific enolase; *p*-value indicates PPD < 5 mm vs. PPD ≥ 5 mm comparison.

## Data Availability

The data supporting the findings of this study are provided within the manuscript and its Appendix A. For any additional requests regarding raw data, please contact the corresponding author.

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
