# Peer review of "Periodontal Bacteria and Outcomes Following Aneurysmal Subarachnoid Hemorrhage: A Prospective Observational Analysis"

_biomedicines, 2025, doi:10.3390/biomedicines14010048_

Round 1
Reviewer 1 Report
Comments and Suggestions for Authors
The manscript addreses a timely and emerging topic—the relationship between periodontal pathogens and outcomes after aneurysmal subarachnoid hemorrhage (aSAH). This mechanistic link is novel and clinically relevant. The study is prospective, integrates periodontal exam, microbiology, biomarkers, and clinical outcomes, and uses culture-validated pathogens. This is a strength. However, there are some gaps that leave room for imprvement, for example methodology clarity, statistical robustness, data interpretation and organization that could be addressed before publication. Parts are somewhat overstated relative to sample size (n=43), confidence intervals are somewhat wide and causal language should certainly be reduced. Improvements to structure, flow, figures, tables and writing would also help polish this submission.
Some specific concerns:
- small sample size / low event numbers
- model overfitting
- too many covariates for the outcome frequency
- potential quasi-complete separation
- Limit covariates in regression (e.g., use only IL-6, hsCRP, PPD ≥5 mm).
- Consider Firth penalized logistic regression.
- Report the number of events per variable (EPV).
- Emphasize exploratory nature rather than strong predictive claims.
- Tone down language implying strong predictive accuracy.
-
Recommendation: clearly state observational nature and avoid causal or mechanistic claims.
- Authors state primary endpoint is DCI and secondary is mRS (outcome)
- But Methods section 2.5 reverses this: “Primary outcomes were mRS at discharge… secondary outcome was DCI”.
- I am somewhat concerned for missclassification bias, so I would suggest the authors offer justification for:
- why full-mouth probing was not feasible (clinical condition, ICU constraints)
- Whether Ramjord teeth reflect generalized periodontitis severity in critically ill patients
-
Consider discussing limitations vs molecular (16S) methods.
- note that culture can miss anaerobes
- CFU interpretation should be discussed cautiously
- Style point: he MALDI-TOF thresholds for species-level ID could be explained more succinctly
-
Why were biomarkers not taken at baseline?
The first 48h after SAH are associated with major variability in IL-6, CRP, S100B.
- Also wondering: were samples collected before any infections developed?
- Table 4 was hard to follow with some misaligned rows, missing headers, unclear CFU comparisons and formatting errors in the version I downloaded.
- Just to clarify, were all aneurysms coiled or were some clipped?
- More discussion on vasospasm precautiopns, and whether these were standArdized.
-
What were your imaging criteria?
-
- Was CT perfusion, TCD and CTA/DSA used?
Vergouwen criteria mentioned but:
- Was CT perfusion, TCD and CTA/DSA used?
-
I could use more detail to help confirm consistency.
-
- Control gorup was not well defined. Eg. were they age and sex matc hed?
- I would rerorganize and clarify the results. here would be a more logical order:
-
Suggested order:
- Cohort characteristics
- Biomarker differences
- Periodontitis severity
- Pathogen spectrum
- Associations with DCI
- Regression modeling
-
Several paragraphs (e.g., lines 378–392) provide textbook information on periodontitis.
Consider condensing to 4–5 sentences to keep focus on the SAH context.
- Grammar and syntax errors and symbol consistency will need to be refined
Author Response
Dear Reviewer 1,
We thank you for the careful evaluation of our manuscript and for the constructive comments provided. We have revised the manuscript accordingly to address all suggestions, which have helped to improve the clarity, accuracy, and overall quality of the work.

Reviewer 2 Report
Comments and Suggestions for Authors
Dear Dr. Pasitka,
I have an honor to review your manuscript named “Exploring the Relationship Between Periodontal Bacteria and Outcomes Following Aneurysmal Subarachnoid Hemorrhage”.
Your work is of undoubted scientific and practical interest, and describes new data regarding previously unexplored pathological relationships between periodontal infection and subarachnoid hemorrhages.
The article is well structured, written in good scientific language, the study design, materials and methods are well represented, the conclusions correspond to the goals and objectives of the work.
While reading the manuscript, I had a few comments that I believe should be addressed/corrected before publishing the article.
- Comments on the style and design of the work.
There are several sentences in the article that look unfinished, possibly due to the omission of individual words. Please carefully correct all typos, in some places they distort the meaning of the work.
In line 68, the sentence appears incomplete.
In lines 73-74, the taxonomic characteristics of bacteria are given in different fonts, please bring the font into a single format (italics), and style (the abbreviated name of one of the bacteria is not indicated).
In line 147, the word neurointensivists sounds professional slang, perhaps you meant experts in intensive neurological medical care.
Line 195 and 197 incorrectly indicate the carbon dioxide formula (most likely, the format was lost when transferring text - you have CO2, not CO2, as is correct. Please correct.
In line 224, the sentence appears incomplete (compare with the similar sentence in line 229).
In line 235, the sentence looks incomplete, the word (measured? valued with?) is missing.
Line 265 omits part of the sentence that loses important meaning. Did you probably mean the same thing below in line 431? Please adjust.
In addition, the values listed in line 264 in the control group are better to draw up a table for clarity, indicating the degree of reliability for each indicator in comparison with the main group.
In Table 1, the term "cerebri anterior" looks incomprehensible - what did you mean? - the anterior regions of the brain? anterior cerebral artery? - please align the term with anatomical terminology.
In Table 2, you have not generally accepted abbreviations, w and w/o, it is better to avoid such abbreviations.
In line 444, the word is missed (patients with deeper pockets - did you mean "deeper periodontal pockets?" Please correct.
- Content Comments
The article ignored an important topic that is widely covered in the context of the systemic influence of oral diseases - oxidative stress. The influence of oxidative stress is not limited only to inflammation and neuroinflammation, about which you write, but also has other, independent mechanisms of the potential relationship between periodontal infection and damage to the central nervous system, I think it is necessary to write at least a few words about this. You casually mention oxidative stress in the context of pathogenic influences of a particular bacteria, in line 77-79, but this is a general mechanism of potential interaction of infection and central nervous system diseases, including endothelial dysfunction of cerebral vessels, violation of the permeability of the blood-brain barrier, violation of redox-regulation pathways, etc.
There are repetitions in the text - in particular, you mention three times (lines 277, 309, 403) that your results showed a significant increase in the concentration of S100B protein, indicating damage to astrogliocytes. Repetitions are superfluous, in addition, this observation, of course, seems to be very important, but needs to be explained or presented in the assumption of possible reasons, which should be expressed in the discussion section.
Finally, in Tables 1 and 3, describing the distribution of indicators in the same study group of 43 patients from different sides, a discrepancy is determined - for example, in Table 1, the maximum value of hsCRP 63.9, IL-6 149 is indicated, but in Table 3 - 41.7 and 55.9, respectively. Apparently, an error occurred in the calculations - after all, we are talking about the same indicators, in the same group of patients, only in Table 1 they are compared in terms of mRS, and in Table 3 in terms of PPD. The average values may change, but the maximum values should not disappear. Please check, correct or provide explanations.
Taking into account the above comments, I believe that your article may be recommended for publication after their elimination.
Author Response
Dear Reviewer 2,
We thank you for the careful evaluation of our manuscript and for the constructive comments provided. We have revised the manuscript accordingly to address all suggestions, which have helped to improve the clarity, accuracy, and overall quality of the work.

Reviewer 3 Report
Comments and Suggestions for Authors
This manuscript investigates the relationship between periodontal pathogens, systemic inflammation, and neurological outcomes, particularly delayed cerebral ischemia in patients with aneurysmal subarachnoid hemorrhage. The topic is original, clinically relevant, and scientifically timely, given the growing recognition of oral–systemic interactions and the role of dysbiosis in neurovascular disease. The authors present prospective clinical data combining periodontal examination, microbiological culture, and biomarker analysis, offering a multidimensional perspective. The manuscript is well structured, the methods are described in sufficient detail, and the authors provide a substantial discussion contextualizing their findings within existing knowledge. However, several issues should be addressed before the manuscript is suitable for publication:
- The logistic regression model identifying PPD ≥ 5 mm as an independent predictor of DCI shows an extremely wide confidence interval (OR 124; 95% CI: 1.4–11359). This indicates instability of the model and possible overfitting given the small sample size (n=43, with 19 DCI events). Authors should either provide model diagnostics (e.g., events-per-variable ratio, multicollinearity assessment), or refrain from claiming independent predictive value and instead describe the finding as exploratory.
- Biomarkers were measured once, between 24–48 hours post-aSAH. This single time point limits interpretation, as inflammatory and brain injury markers can have dynamic trajectories. Clarify in the Discussion that single timepoint measurement limits the ability to assess temporal associations with DCI, and that biomarker elevations may reflect initial hemorrhage severity rather than periodontal factors.
- The authors rely only on culture-based methods, which detect a fraction of the oral microbiota. Discuss this limitation explicitly and outline how metagenomic or 16S rRNA sequencing could capture uncultivable taxa relevant to neurovascular risk.
- The healthy control group (n=10) is used only for biomarker comparison, yet periodontal sampling was not performed. Authors should clarify why periodontal assessment was not obtained from controls, and whether matching or adjustment for oral hygiene variables was considered.
- While the idea of periodontal screening in aSAH patients is interesting, the manuscript currently overstates its clinical applicability. Authors should emphasize that findings are preliminary, no interventional data show that periodontal treatment modifies DCI risk, and that clinical translation requires validation in larger multicenter cohorts.
- In the Introduction provide a more balanced overview of existing evidence linking oral pathogens to vascular inflammation. A few statements are repetitive and could be condensed. Also some additional references may be needed to support claims about endothelial dysfunction and neuroinflammation mechanisms.
- In Methods periodontal probing description is thorough, but please clarify inter-examiner reliability (was only one examiner used?), and how missing Ramfjord teeth were handled.
- In Methods microbiological processing is detailed, however indicate whether anaerobic conditions were validated (e.g., use of indicators). Moreover, mention reproducibility or quality control measures for MALDI-TOF.
- In Discussion several paragraphs repeat similar concepts (e.g., systemic inflammation mechanisms). Consider tightening to improve focus. Also, expand briefly on why NSE was associated with periodontal status while S100B was not.
Author Response
Dear Reviewer 3,
We thank you for the careful evaluation of our manuscript and for the constructive comments provided. We have revised the manuscript accordingly to address all suggestions, which have helped to improve the clarity, accuracy, and overall quality of the work.

Round 2
Reviewer 3 Report
Comments and Suggestions for Authors
The manuscript has been revised as suggested. I have no other comments.